# Correlations between Dental Age, Skeletal Age, and Mandibular Morphologic Index Changes in Turkish Children in Eastern Anatolia and Their Chronological Age during the Pubertal Growth Spurt Period: A Cross-Sectional Study

**DOI:** 10.3390/diagnostics14090887

**Published:** 2024-04-24

**Authors:** Fatma Saraç, Büşra Baydemir Kılınç, Periş Çelikel, Murat Büyüksefil, Muhammet Burak Yazıcı, Sera Şimşek Derelioğlu

**Affiliations:** 1Department of Pedodontics, Faculty of Dentistry, Atatürk University, 25240 Erzurum, Türkiye; peris.celikel@atauni.edu.tr (P.Ç.); muratb@atauni.edu.tr (M.B.); yazici.m@atauni.edu.tr (M.B.Y.); simseksera@gmail.com (S.Ş.D.); 2Morgue Department, Council of Forensic Medicine, Erzurum Branch, 25010 Erzurum, Türkiye; busrabaydemirk@gmail.com

**Keywords:** forensic anthropology, forensic dentistry, dental age, bone age, chronological age, pubertal growth spurt period

## Abstract

In age determination, different methods aiming to obtain the closest result to chronological age have been investigated so far. The most commonly used one among these is the radiological method, which is usually used to evaluate the developmental stages of wrist bones or teeth. In our study, we assessed bone age estimations using the Gilsanz–Ratib atlas (GRA), which has recently become commonly used for children aged 9 to 15 years; evaluated the dental age, determined with Cameriere’s European method; conducted morphometric measurements of the mandibular bone; and then examined their relationships with chronological age. The results of our study reveal that, in children during the puberty growth spurt, Cameriere’s EU formula might have higher accuracy in estimating chronological age in younger age groups, while the GRA might be more accurate for older ages. Additionally, we conclude that of the mandibular morphometric measurements, condylar height and tangential ramus height show strong positive correlations with age. As a result, we conclude that the morphometric measurements evaluated in the present study can be used as auxiliary methods in forensic anthropology and forensic dentistry.

## 1. Introduction

Age estimation is of the utmost importance for forensic use in anthropology and dentistry, in which different dental and skeletal maturity assessment methods are frequently used. Various approaches, such as morphological, histological–biochemical, radiomorphological, and radiometric techniques, have been evaluated in age determination [1,2,3,4]. The radiological method, which is the one most commonly used among all, examines the phases of skeletal maturity and dental development. The radiological method often involves hand and wrist X-ray imaging for bone age determination and orthopantomography (OPG) for dental age estimation. Both radiologic techniques are very advantageous in the age determination process since they are user friendly and irradiate low-radiation doses [3].

The Gilsanz–Ratib atlas (GRA) has recently become very popular for visually reviewing the morphological changes in hand–wrist X-rays. The GRA provides a collection of reference digital hand–wrist radiographs showing different levels of maturity in the ossification centers in hand and wrist bones peculiar to every gender and age groups. Unlike the Greulich–Pyle atlas, in the GRA a single standard hand–wrist X-ray graph of a particular individual is not accepted as a reference image. Rather, hybrid computer-generated images (CGIs) that compound several digital hand–wrist radiographs are artificially formed to more accurately represent the idealized image of bone maturation [5]. To take advantage of this, we used the GRA in our study to estimate the skeletal age of children in the puberty growth spurt period.

Several modalities assessing the stages of dental maturity by OPGs have been generated and used in different populations [6,7]. For acceptability, these age estimation techniques should provide approximate scores to the chronological age and provide consistent results in repeated evaluations in the same individual [1]. However, there is no universal method for dental age determination yet. Dental age assessment approaches based on calcification may be specific to the population for which they were developed. Different growth and development phases seen in different communities require a reliability analysis in different populations [6,8,9]. It has been reported that one of those methods, Cameriere’s EU age estimation formula, as presented by Camerire’s et al. for European and neighboring countries, could be used in different regions of Türkiye [10,11]. On this basis, Cameriere’s EU method was selected in the present study to assess the correlation between the dental and chronological ages of children in the puberty growth spurt period living in the eastern part of Türkiye.

It is an advantage for age estimation that mandibular morphology evolves with age, gender, occlusal status, and muscular functions and differs from other facial bones during growth and development. Moreover, the usability of mandibular morphology in age determination has previously been reported, because there is a dense layer of cortical bone and a consequent durability against traumas [12,13]. Additionally, mandible grows in parallel with dental maturity. All of these characteristics show that mandibular measurements could be used for dental age estimation. As it reveals the morphological characteristics of populations, the assessment of bone structure is also very important for guiding anthropological research [14]. Ethnicity is also another parameter affecting mandibular growth and morphometrical measurements [15]. 

The present study was carried out to create a supplementary reference for use in judicial cases. Late birth registrations, which are common in Eastern Anatolia, result in the necessity of official age determinations in legal cases [16]. Thus, physical, oral, and radiological examinations are used to determine child marriages and criminal liability in Türkiye. According to the Turkish penal code, children under the age of twelve have no criminal liability, and safety measures exclusive to minors apply to such children who have committed crimes. The criminal liability of children aged 12 to 15 years is determined through forensic psychiatric examinations [17]. Children above 15 years of age are regarded as fully responsible for criminal actions. Thus, in our study, we evaluated 9–15-year-old children to contribute to forensic age determination efforts.

The present study aimed to evaluate the correlation between the dental age scores estimated using the GRA, Cameriere’s EU formula, mandibular morphologic measurements and the chronological ages of the children living in the eastern part of Türkiye during their pubertal growth spurt period.

## 2. Material and Method

### 2.1. Patient Selection

The present study involved 240 children (120 girls and 120 boys), aged 9–15 years, admitted to the Department of Pedodontics at the Ataturk University Faculty of Dentistry with previously taken OPGs (ProMax®, Planmeca Oy, Asentajenkatu 6, 00880 Helsinki, Finland) and hand–wrist X-rays (ProMax®, Planmeca Oy, Asentajenkatu 6, 00880 Helsinki, Finland).

### 2.2. Inclusion Criteria

Living in Eastern Anatolia;Being right-handed;Having no systemic disorders;Having X-ray images clearly revealing the bone structures;Having no muscular dystrophy, congenital anomalies, or previous history of trauma at the related areas that can adversely affect the growth of hand–wrist bones.

### 2.3. Exclusion Criteria

Systemic diseases in the anamneses;Previous history of orthodontic treatment or appliances;Hypodontia, extracted or missing, with the exception of permanent third molars, deep carious lesions, restorations, apical lesions, and root canal therapy in the left mandible.

Hand–wrist X-rays and OPGs were chosen from patients’ previous radiographs, which were taken for indications of orthodontic treatments and routine dental treatments. There was no time interval greater than one month between the dates of patients’ hand–wrist X-rays and OPGs.

### 2.4. Skeletal Age Determination

Bone age was determined using the discrete reference left hand X-ray images for boys and girls in the GRA by matching them with participants’ left hand X-rays and choosing the most appropriate image.

### 2.5. Chronological Age Determination

Patients’ chronological ages were determined as the “dates of the OPGs-official birthdates/365.25” in the decimal system in Excel 2016. Furthermore, the chronological age and gender of the patients were recorded.

### 2.6. Dental Age Determination

The dental ages of the patients were calculated with Cameriere’s EU formula [18] using the measurements on their OPGs. For the mandibular measurements, all digital OPGs were loaded into computer image-processing software (ImageJ 1.53 V. National Institutes of Health and the Laboratory for Optical and Computational Instrumentation-LOCI, University of Wisconsin). The number of mature teeth with closed apices were counted, in accordance with Cameriere et al. [18], and abbreviated as N_0_. For immature teeth with open apices, the distance between the inner sides of the apex was measured in single-rooted teeth (Ai, i = 1, …, 5), and the sum of the distances between the inner sides of the two apices was calculated (Ai, i = 6, 7) in multiple-rooted teeth. Measurements were divided by the tooth length (Li, i = 1, …, 7) to minimize possible magnification and angulation errors in the OPGs. For the second premolars, the X_5_ value was calculated by dividing A_5_ by L_5_. And, finally, dental age was assessed by placing the obtained scores in Cameriere’s EU formula (DA = 8.387 + 0.282 × g − 1.692 × X_5_ + 0.835 × N_0_ − 0.116 × s − 0.139 × s × N_0_) (Gender [G] = 1 for boys, 0 for girls, and s = ∑ A_i_/L_i_) (Figure 1).

### 2.7. Morphometric Measurements

The maximum ramus width (MaxRW), minimum ramus width (MinRW), condylar height (ConH), coronoid height (CorH), and tangential ramus height (RHt) of the mandible were measured in this study. For intrarater consistency, each measurement was repeated three times, using Image 1.49 V, at one-week intervals, and the average of the three measurements was recorded. Measurements were taken using the radiographic images, which were compatible with the inclusion criteria, as shown in Figure 2.

### 2.8. Statistical Analysis

In the present study, descriptive statistics are expressed as the count, mean, standard deviation, minimum, and maximum. The reliability of the study scales was statistically tested. First, the Shapiro–Wilk test was used to assess the assumption of normality. Outliers were identified with the boxplot method. The Whitney U test was performed to compare the differences among the means of independent variables in the groups without assuming normality. The Kruskal–Wallis test was used to compare the means of two or more independent groups without normal distribution. A post hoc Bonferroni analysis was conducted to indicate significant differences among the groups. The relationship between two continuous variables without normal distribution was assessed with the Spearman’s rank-order correlation test. All data were analyzed with IBM ^®^ SPSS^®^ 25.

## 3. Results

This study consisted of 240 participants (120 girls and 120 boys) at the chronological age of 9 to 15 years. As a result of the analyses conducted to compare the measurements, a statistically significant difference was found between the scores of the chronological, dental, and skeletal ages (*p* < 0.05). The chronological age scores were observed to be higher than the dental and skeletal ages (*p* < 0.001 and *p* < 0.001), whereas the skeletal age scores were higher than the dental ages (*p* < 0.001) (Table 1, Figure 3).

The relationships between the mean chronological, skeletal, and dental age scores in girls and boys by age group are given in Table 2 and Figure 4. In the 9-, 10-, and 11-year-old girls’ patients, the higher chronological age scores than the bone age were statistically significant (*p* = 0.009, *p* = 0.003, and *p* < 0.001). The higher scores of chronological age than the bone age seen in the 10- and 12-year-old girls’ patient groups were also found to be statistically significant (*p* < 0.001 and *p* = 0.001). In 13- and 14-year-old female patients, the chronological and skeletal age scores were significantly higher than their dental age scores (*p* < 0.001 and *p* < 0.001).

The chronological age scores observed to be greater than skeletal age in 10-year-old boys were statistically significant (*p* = 0.008). The 11-year-old male participants’ chronological ages were significantly higher than their dental and skeletal ages (*p* < 0.001 and *p* < 0.001). In the 12-year-old boys’ patient group, the chronological and skeletal age scores were significantly higher than the dental age scores (*p* = 0.022 and *p* < 0.001). The chronological and skeletal age scores of the 13-year-old boys were observed to be significantly higher than their dental ages (*p* = 0.028 and *p* < 0.001). The higher scores of chronological age than the bone and dental age scores in the 14-year-old boys’ patient group were also statistically significant (*p* < 0.001 and *p* < 0.001).

The boxplot method was used for measuring mandibular morphometric scores and for identifying the outliers. The analyses revealed outliers both in the coronoid and tangential ramus height measurements, and these outliers were included in the analyses’ results to avoid bias.

In the present study, the data analyses revealed a statistically significant and strong positive relationship between the condylar and tangential ramus heights and chronological, dental, and skeletal age scores (*p* < 0.05). We observed a significant moderate positive correlation between the chronological, dental, and skeletal age scores and scores of other morphometric measurements (*p* < 0.05) (Table 3). 

The statistical analyses showed significant and moderate/strong positive correlations between the chronological, dental, and skeletal age scores and the condylar and tangential ramus heights in the girls’ patient group (*p* < 0.05). There were statistically significant, moderate positive correlations between the scores of the chronological, dental, and skeletal ages and the coronoid heights and maximum ramus widths (*p* < 0.05). Significant, weak positive relationships were found between the minimum ramus widths and the chronological, dental, and skeletal ages (*p* < 0.05).

We observed significant, strong positive correlations between chronological, and dental, skeletal ages and coronoid height scores in the boys’ patient group (*p* < 0.05). whereas there were statistically significant, moderate/strong positive relationships between the tangential ramus heights and the chronological, dental, and skeletal ages (*p* < 0.05). Moreover, significant, moderate positive correlations between the chronological, dental, and skeletal ages and the coronoid height and maximum/minimum ramus width scores were seen among the male participants (*p* < 0.05) (Table 4).

## 4. Discussion

In age estimation, radiological methodology is a more frequently chosen approach than histological and biochemical analysis techniques, since it is more economical and convenient. It was reported that using multiple scientifically proven methods in forensic identification and age determination processes achieve more reliable results [19,20]. Evaluating skeletal and dental maturity is a preferred methodology for age estimation. Chronological age can be determined by assessing dental and mandibular maturity [21].

In this study, the morphological scores of the mandible such as, maximum and minimum ramus widths, tangential ramus height, condylar height, and coronoid height, were measured. Additionally, another dental age determination approach, Cameriere’s EU formula, which was used for assessing the Turkish children [10,11], as well as the GRA, a recently popularized forensic skeletal age estimation technique, were also used. The present study examined the correlations between the dental and skeletal ages of children, as determined by different methods, in the pubertal growth spurt period and their chronological ages by considering gender and regional variations.

The assessment of bone morphology is significant, since it demonstrates the morphological characteristics of populations and guides anthropological research [14,15]. It has been reported that ethnic differences influence mandibular maturity and morphometrical measurements. In their research, conducted utilizing mandibular morphometric measurements taken using digital OPGs, Bhuyan et al. [22] illustrated a correlation between chronological age and mandibular morphometric values.

Similarly, we also found positive correlations between all patients’ condylar and tangential ramus heights and their chronologic, dental, and skeletal age scores. Furthermore, in the gender-based analysis, statistically significant positive correlations were also observed between the condylar and tangential ramus height measurements and the chronological, dental, and skeletal age scores of both girls and boys. Motawei et al. [15] reported that there was a strong correlation between the morphology of the mandible, especially the mandibular ramus, and chronological age. The results of our study reveal that mandibular morphometry might be useful in age estimation. 

Several dental age estimation approaches are based on the extent of the calcifications in permanent teeth detected through radiographic evaluations [9,11]. Of these methodologies, one of the most widely used is the eight-stage method, introduced by Demirjian and Goldstein. However, these techniques have been used in different populations and resulted in mismatches between dental and chronological age scores. This situation gave researchers the opportunity to recommend population-based, standard age-estimation modalities [23,24]. In fact, a study comparing dental maturity in Central and Eastern Anatolia reported that the climatic conditions had an effect on the dental maturity and that dental development is relatively slower in the eastern regions of Türkiye [25]. Additionally, in their research carried out in Eastern Türkiye, using the Demirjian system, Celikoglu et al. [26] reported that this approach was not accurate for use in eastern Turkish societies, since dental age scores were found to be significantly overestimated compared to chronological age. Although we did not research the adoptability of Cameriere’s EU method for child populations in the eastern parts of Türkiye, it was previously reported to be suitable for use in different regions [10,11]. 

Thus, we preferred to use Cameriere’s EU approach in our study. The distributions of chronological and dental age by gender and age groups revealed that the chronological age scores were higher than the dental age scores among both girls and boys aged 12, 13, and 14 years. We concluded that Cameriere’s EU formula achieved the most accurate results in 9-year-old girls and 9–10-year-old boys living in the eastern part of Türkiye, but the error rates increased with age. Reinforcing the results of this study, Nolla’s and Cameriere’s methods were compared in a previous study conducted in Turkish children with a mean age of 9–14 years; Cameriere’s EU approach was suggested for use in evaluating the younger kids, whereas Nolla’s method was for the children in the older age groups [10]. It has been thought that the decline in the accuracy of the estimations were caused by the completion of dental maturation [11,27]. Also seen in the present study, as chronological age increased, the gap between the dental ages estimated with Cameriere’s EU formula and chronological ages expanded with a growing rate of error. Bearing this in mind, it may be said that Cameriere’s EU technique works relatively better at the beginning of puberty, but the failure rate increase with age.

The Eastern Anatolia Region has, generally, fewer sunny days due to the long cold winter season. A previous study carried out in the same region and city remarked that a significant part, as much as 72%, of children aged 13–17 years old suffered from vitamin D insufficiency, while 17.7% of them experienced deficiency in vitamin D, which is an essential hormone for the intestinal absorption of calcium (Ca), magnesium (Mg), and phosphorous (P) required for optimum bone, dentin, and enamel mineralization [28]. The relationship between delayed eruption and vitamin D deficiency has also been shown in recent studies [29,30,31]. Although we did not test the participants’ blood vitamin D levels, we thought that their lower scores of dental age than chronological age was associated with the levels of vitamin D.

Human skeletal maturity, also called the bone age, represents the point of physiological development that reflects the size, shape and degree of bone mineralization [4]. The most frequently used diagnostic tool for skeletal age estimation in children is hand–wrist X-ray imaging. In this technique, bone age is determined on the basis of the extent of the ossification of the epiphyseal bone plate in the hand. In this methodology, the degree of skeletal maturity is regarded as the bone age, which is determined by comparing the radiographic images of reference individuals [5]. The GRA is a newly popularized age determination tool for evaluating the morphological variables in digital- or film-based hand–wrist X-ray images [5]. The latest electronic version of this atlas consists of artificial images generated by integrating several radiographs, which illustrate different maturation stages of ossification centers in the gender and age specific hand and wrist X-rays [32]. Thus, highly illustrative, idealized artificial radiographic images were created. Since there was no software designed for measuring other age estimation parameters, they were measured manually on the radiographs by different researchers in the present study. And in lieu of the digital atlas, a hard copy form was used for verifying standardization and consistency among the studies.

The scores of the GRA-based dental age determinations were found to be lower than the chorological age scores in 9–11-year-old girls and 10- and 11-year-old boys. Nevertheless, in 12–14-year-old girls, similar chronological and dental age scores were obtained. The chronological and dental age scores were observed to be similar in 12- and 13-year-old male patients. The study data revealed that for the children living in the Eastern Anatolia, error rates in the GRA-based age determination approach might be high at early puberty but relatively lower in the following periods of puberty. In our study, we presumed that dental age scores found to be closer to chronological age in the older group of patients might be associated with the increased bone density and maturity at the onset of the pubertal growth spurt. Since there were only a limited number of studies in the literature evaluating the outcomes of the GRA-based age estimations carried out for the children in the Eastern Anatolia, we thought that the results of our study were important.

The present retrospective study has some limitations that should be accounted for In the future researches; the most prominent one is the lack of a comprehensive assessment of the environmental factors, which may affect the participants’ growth and development such as, socioeconomic status, dietary habits, hemodynamic and biochemical parameters, and way of life. Additionally, the correlation between chronological and dental age should be assessed with different methods in a larger study population of the pubertal children. Further studies will help developing accurate and reliable age estimation techniques in different societies.

## 5. Conclusions

The results of the present study illustrate that for pubertal children living in Eastern Anatolia, Cameriere’s EU formula might be more accurate in estimating chronological age of younger ones, whereas the accuracy of the GRA technique might be higher for older ages. We also conclude that changes in the condylar and ramus height measurements were correlated with the chronological age and these might be used as subsidiary morphometric techniques in forensic anthropology and dentistry. We believe that novel age estimation modalities should be adopted or current methods should be modified for forensic investigations in the region where the present study was carried out.

## Figures and Tables

**Figure 1 diagnostics-14-00887-f001:**
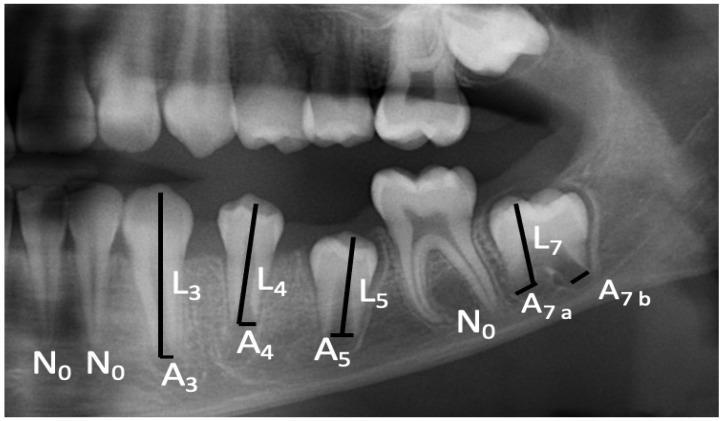
Cropped OPG image showing the dental age estimation method used.

**Figure 2 diagnostics-14-00887-f002:**
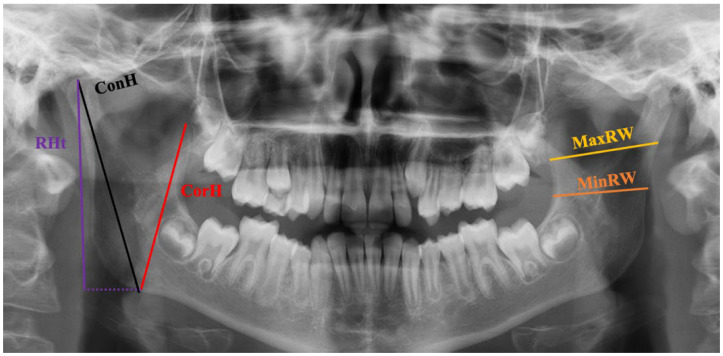
OPG image illustrating the morphometric measurements.

**Figure 3 diagnostics-14-00887-f003:**
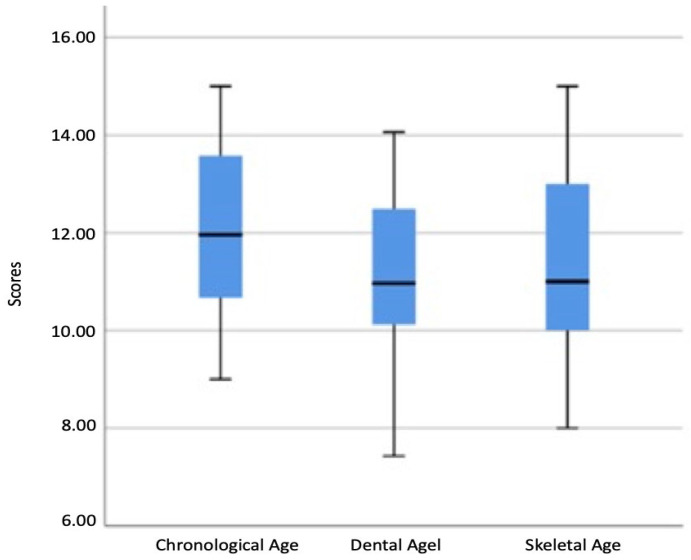
The distribution of the chronological, dental, and bone age scores.

**Figure 4 diagnostics-14-00887-f004:**
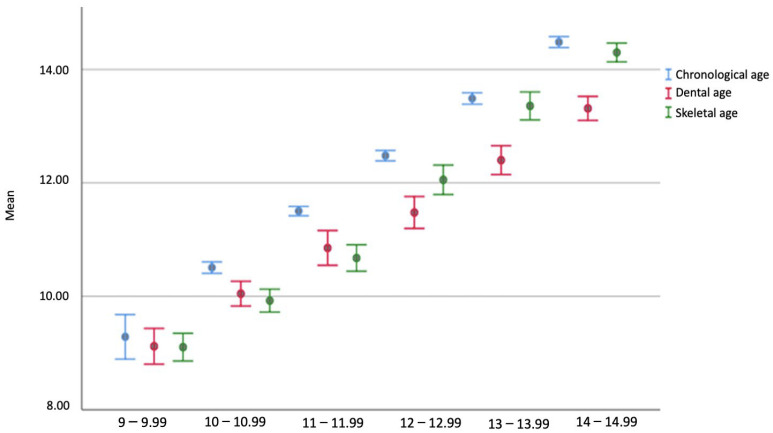
Boxplot graph of the chronological, dental, and bone age scores by age group.

**Table 1 diagnostics-14-00887-t001:** Comparison of the chronological, dental, and skeletal age scores.

	Min.–Max.	Mean ± S.D. (M.)	*p*
Chronological age	9.0–15.0	11.9 ± 1.8 (11.9)	<0.001 *
Dental age	7.4–14.0	11.2 ± 1.6 (10.9)	
Skeletal age	5.0–15.0	11.5 ± 1.9 (11.0)	

S.D.: standard deviation; M: median. * *p* < 0.05.

**Table 2 diagnostics-14-00887-t002:** Comparison of the chronological, dental, and skeletal age scores by the age and gender groups.

Age Group	Gender		Min.–Max.	Mean ± S.D. (M.)	*p*
9–9.99 years	Girl	Chronological age	9.0–9.9	9.4 ± 0.2 (9.4)	0.008 *
		Dental age	7.4–10.2	9 ± 0.91 (9)	
		Skeletal age	8–10	9.1 ± 0.4 (9)	
	Boy	Chronological age	9.0–9.9	9.0 ± 1.7 (9.4)	0.056
		Dental age	7.8–11.5	9.2 ± 1.0 (8.7)	
		Skeletal age	8–12	9.1 ± 0.9 (9)	
10–10.99 years	Girl	Chronological age	10.0–10.9	10.6 ± 0.2 (10.7)	0.000 *
		Dental age	8.9–12.1	10.1 ± 0.6 (10)	
		Skeletal age	9–11	9.7 ± 0.6 (10)	
	Boy	Chronological age	10–10.9	10.4 ± 0.3 (10.3)	0.008 *
		Dental age	8.6–10.9	10 ± 0.7 (10.1)	
		Skeletal age	9–11	10.0 ± 0.5 (10)	
11–11.99 years	Girl	Chronological age	11.0–11.9	11.4 ± 0.2 (11.4)	0.001 *
		Dental age	8.7–12.9	11.1 ± 1.1 (11.2)	
		Skeletal age	9–13	10.6 ± 0.8 (11)	
	Boy	Chronological age	11.0–11.9	11.5 ± 0.2 (11.5)	0.000 *
		Dental age	8.6–11.5	10.4 ± 0.6 (10.5)	
		Skeletal age	10–12	10.6 ± 0.5 (11)	
12–12.99 years	Girl	Chronological age	12.0–12.9	12.0 ± 0.3 (12.5)	0.001 *
		Dental age	10.0–12.8	11.7 ± 0.9 (11.9)	
		Skeletal age	10–14	12.1 ± 0.8 (12)	
	Boy	Chronological age	12–12.9	12.4 ± 0.2 (12.5)	0.000 *
		Dental age	10.0–12.5	11.2 ± 0.7 (10.8)	
		Skeletal age	11–13	11.9 ± 0.7 (12)	
13–13.99 years	Girl	Chronological age	13–13.9	13.4 ± 0.3 (13.5)	0.000 *
		Dental age	11.3–13.6	12.4 ± 0.6 (12.2)	
		Skeletal age	12–15	13.4 ± 0.8 (13)	
	Boy	Chronological age	13–13.9	13.5 ± 0.3 (13.5)	0.000 *
		Dental age	10.7–14.0	12.4 ± 0.9 (12.4)	
		Skeletal age	11–15	13.2 ± 0.7 (13)	
14–14.99 years	Girl	Chronological age	14–14.9	14.4 ± 0.2 (14.5)	0.000 *
		Dental age	12.0–13.6	13.1 ± 0.6 (13.6)	
		Skeletal age	13–15	14.4 ± 0.6 (14)	
	Boy	Chronological age	14–14.9	14.5 ± 0.3 (14.5)	0.000 *
		Dental age	12.4–14.0	13.4 ± 0.6 (14)	
		Skeletal age	14–15	14.1 ± 0.3 (14)	

S.D.: standard deviation; M: median. * *p* < 0.05.

**Table 3 diagnostics-14-00887-t003:** Correlation between the scores of the chronological, dental, and skeletal ages and the morphometric measurements.

		Chronological Age	Dental Age	Skeletal Age
Condylar height	r	0.758	0.753	0.802
	*p*	<0.001	<0.001	<0.001
Coronoid height	r	0.620	0.652	0.671
	*p*	<0.001	<0.001	<0.001
Ramus height	r	0.700	0.712	0.763
	*p*	<0.001	<0.001	<0.001
Maximum ramus height	r	0.579	0.561	0.580
	*p*	<0.001	<0.001	<0.001
Minimum ramus height	r	0.325	0.335	0.333
	*p*	<0.001	<0.001	<0.001

r: Correlation coefficient.

**Table 4 diagnostics-14-00887-t004:** Relationship between the scores of the chronological, dental, and skeletal ages and the mandibular morphometric measurements by gender.

		Girl			Boy		
		Chronological Age	Dental Age	Skeletal Age	Chronological Age	Dental Age	Skeletal Age
Condylar height	r	0.749	0.688	0.773	0.771	0.807	0.841
	*p*	<0.001	<0.001	<0.001	<0.001	<0.001	<0.001
Coronoid height	r	0.605	0.618	0.654	0.630	0.661	0.692
	*p*	<0.001	<0.001	<0.001	<0.001	<0.001	<0.001
Tangential ramus height	r	0.717	0.648	0.753	0.694	0.761	0.786
	*p*	<0.001	<0.001	<0.001	<0.001	<0.001	<0.001
Maximum ramus width	r	0.525	0.496	0.518	0.644	0.656	0.672
	*p*	<0.001	<0.001	<0.001	<0.001	<0.001	<0.001
Minimum ramus width	r	0.251	0.230	0.240	0.385	0.453	0.445
	*p*	0.006	0.011	0.008	<0.001	<0.001	<0.001

r: Correlation coefficient.

## Data Availability

The data presented in this study are available upon request from the corresponding author.

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
