# Peer review of "Correlations between Dental Age, Skeletal Age, and Mandibular Morphologic Index Changes in Turkish Children in Eastern Anatolia and Their Chronological Age during the Pubertal Growth Spurt Period: A Cross-Sectional Study"

_diagnostics, 2024, doi:10.3390/diagnostics14090887_

Round 1
Reviewer 1 Report
Comments and Suggestions for Authors
1. Some typos and writing need to be corrected, for example:
a) Line 2: Skelatal should be Skeletal
b) Keywords: forensic dentistry; dental age
c) Line 29: (OPG)
d) Line 239: Although
e) some spacing existed
2. Are there any differences between Eastern and other regions in Turkiye? Please include a bit of explanation in the discussion part.
3. What is the purpose of determining the age of the 9-15-year group? Is there any relevance to the forensic investigation as mentioned in the conclusion?
Comments on the Quality of English LanguageProfessional English editing should be included.
Author Response
- Grammar errors and misspellings were corrected
- Amendments explaining regional differences / variations were made in the discussion section (Lines 364-367)
- The reason why we chose 9-15 years of age range as a basis for the study and its forensic /judicial importance was explained in the introduction section (Lines 77-87)
- Language was revised.
Reviewer 2 Report
Comments and Suggestions for Authors
The article is well written, focused, well-presented. It might make sense to include the geogaphical limitation in the title. The main thing I am missing is the justification, why this age frame is so important to understand in such a detail.
Some minor issues are in the file.

Author Response
- Minor amendments were done. (Please check the annex)
- Title was rewritten in accordance with the reviewer’s recommendation.
- The reason why we chose 9-15 years of age range as a basis for the study was explained in the introduction section (Lines. 77-87)